# SemCLIP: A Semantic Memory-Aligned Vision Language Model

## Abstract

Vision-language models (VLM) bring image and textual representations close together in a joint embedding space, which is useful for tagging and retrieval from content stores. However such associations are not very stable in that a synonymous textual query does not retrieve the same set of images or with a high degree of overlap. This is due to the absence of linkages between semantically related concepts in vision-language models. In contrast, the episodic memory store in the brain has linkages to the semantic conceptual memory subsystem which helps in both the formation and recall of memories. In this paper, we exploit this paradigm to link a VLM to a semantic memory thereby producing a new semantic vision-language model called SemCLIP. Specifically, we develop a semantic memory model for the language of object-naming nouns reflecting their semantic similarity. We then link a vision language model to the semantic memory model through a semantic alignment transform. This leads to a richer and more stable understanding of the concepts by bringing synonymous visual concepts and their associated images closer. Both the semantic memory model and the alignment transform can be learned from word knowledge sources thus avoiding large-scale retraining of VLMs from real-world image-text pairs. The resulting model is shown to outperform existing embedding models for semantic similarity and downstream tasks of retrieval on multiple datasets.

## 1 Introduction

More and more enterprises are opting for content stores for managing large collections of photos, video, audio, and documents [22, 2, 19]. In these, the content is stored as vectors, associated with textual tags in a vision-language model (VLM) and retrieved with vectors formed from textual queries [7, 15, 6, 29, 8]. However such associations are not very stable in that a synonymous textual query does not retrieve the same set of images or even those with a high degree of overlap. Figure 1 illustrates this problem, showing examples of the top 5 retrieved images from sets of similar queries prompted by "Images of X" where X is the phrase on top of each column. In Figure 1(a)-(b), synonymous terms "hamper" and "basket" retrieve different top 5 matches. This problem is also seen when more context is available as in the queries of Figure 1(c)-(d) where more terms are replaced by their synonymous phrases (overcoat→coat, frock→gown) or multiple objects are queried in different order (Figure 1(e)-(g)).

This instability is an inherent limitation of the underlying textual embeddings used to build the VLMs which use self-supervised method of inferring meaning similarity based on use context. As an example, the fraction of the English language nouns that are near their synonyms in existing textual and VLM embeddings is shown in Table 1. Here 70,000 single sense nouns from the WordNet thesaurus [4] were projected into textual and VLM embedding spaces and their synonyms in the top

Submitted to 39th Conference on Neural Information Processing Systems (NeurIPS 2025) UniReps Workshop. Do not distribute.

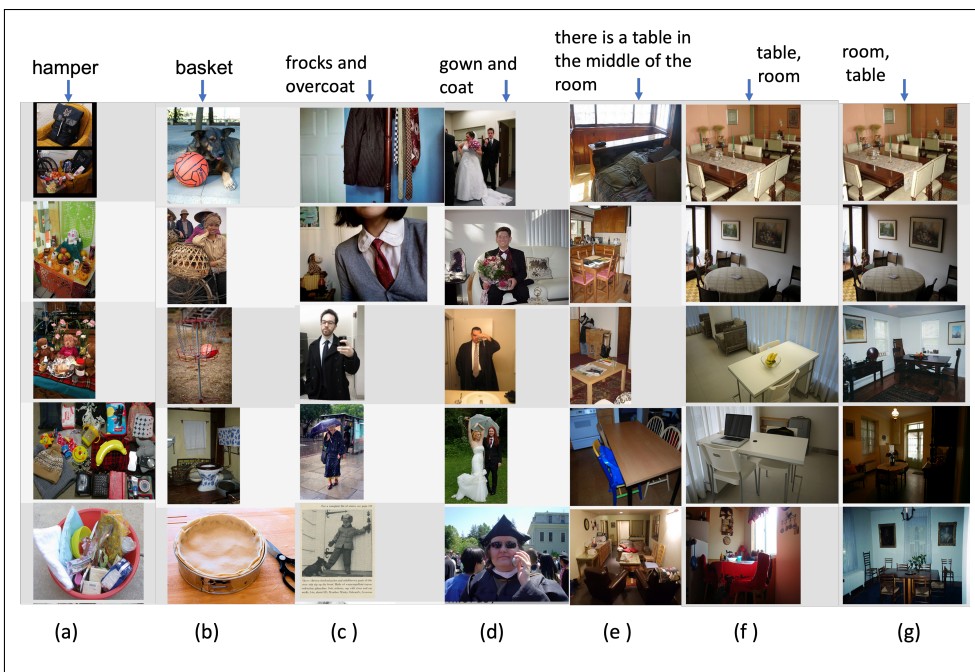

Figure 1: Illustration of retrieval instability to synonymous phrases in vision language models (CLIP[15]). (a)-(b) Two synonymous queries retrieving different results. (c)-(d) Multi-word queries with synonymous replaced terms. (e)-(g) Effect of order of terms in queries.

10 neighbors were noted. As can be seen from this table, there is at best a 50% overlap with their synonyms indicating missing linkages to their semantically related concepts.

In contrast, the formation and recall of memories in the brain does utilize the linkage between episodic and semantic memory systems. It is now widely acknowledged that these two forms of memory interact during both encoding and retrieval[23]. Extensive behavioral, lesion, and functional imaging studies have demonstrated the existence of a semantic memory system in brain for organizing and interpreting episodic memories distributed throughout the cortex for representing distinct object categories such as people, animals, and tools[18]. Neurological evidence also exists for the inter-dependencies between semantic and episodic memory[5]. Semantic memory facilitates the acquisition of new episodic memories, and episodic memory facilitates the addition of new information to the semantic store. Similarly, episodic memories facilitate the retrieval of information from semantic memory, and semantic memories are the basic material from which complex and detailed episodic memories are constructed[5]. In this paper, we draw inspiration from this human memory paradigm

to develop a new semantically-guided vision language model called SemCLIP. In doing so, we make three novel contributions. (1) First, we develop a new textual embedding (STE) as a semantic memory reflecting the similarity relations between all object-naming nouns in the English language. It is learned using multi-label supervised contrastive learning on data derived from a constrained traversal of the WordNet thesaurus to cover all English language nouns and their semantically similar concepts. (2) Next, we develop a semantic alignment transform to link VLM embeddings to the semantic memory. The alignment transform is a custom-designed neural network trained to map between textual embeddings projected in VLMS to those in semantic memory. It is learned from a vocabulary representing the diversity of language use across different domains as captured in the captions of multiple datasets. Image embeddings in the VLM can then be aligned using the transform of their nearest textual embedding. (3) Finally, we provide the training dataset for the STE embedding consisting of 114,000 linguists-curated similarity lists of words and over 600,000 pairs of synonyms terms derived from these similarity lists as contributions to open source, which can be valuable for other researchers for many downstream NLP tasks.

The SemCLIP approach leads to a richer and more stable understanding of the concepts by bringing synonymous visual concepts and their associated images closer. Furthermore, the semantic memory

model and the alignment transform can be learned from textual knowledge sources thus avoiding large-scale retraining of VLMs from real-world image-text pairs. The resulting model is shown to outperform existing embedding models for semantic similarity and downstream tasks of retrieval on multiple datasets.

## 2 Related Work

To our knowledge, the paradigm of episodic-semantic memory interactions has not been used before to generate vision-language models. Further, insights into the stability aspects of retrieval or the limitations of textual embeddings in influencing VLM models haven't been addressed in detail before. Other prior works, however, have pointed to issues with text to image retrieval and image tagging with CLIP with many variants of CLIP developed to address issues such as semantic inconsistencies[28], and augmenting CLIP training with knowledge graphs to allow a better understanding of the semantics in queries[14]. StructureCLIP [9] mentions that existing methods often perform poorly on image-text matching tasks that require a detailed semantic understanding of the text and recommended augmenting VLMs with scene graphs composed of objects, attributes, and relations. BLIP [13] and its variants are unified vision-language models using a multimodal mixture of encoder-decoder architectures trained with a language modeling loss to generate better captions given images. Sigmoid Loss for Language-Image Pre-training (SigLIP and SigLIP2) [30] introduced a pairwise sigmoid loss allowing the method to solely focus on the individual image-text pairs. The need for modeling coarse and fine-grained concepts was also emphasized in a recent work [27]. In all improvements proposed for VLM models such as CLIP, the textual embeddings were nevertheless still based on transformer models which infer semantic similarity primarily by use context. In our approach, we achieve the desired improvements by focusing at a different end, namely, improving the semantics in textual embedding and using an alignment transform to project from the original CLIP model to form a new space of semantically connected words and images.

## 3 Developing a semantic memory model for VLMs

Various knowledge graphs and thesaurus exist to capture different types of relationships between word concepts such as Wordnet[4], ConceptNet[20]. In Wordnet[4], lingusits curated related terms and defined synonyms, generalizations and specializations of concepts. Attempts have been made to use these thesauruses in conjunction with word embeddings acquired from distributional semantics such as Word2Vec or Glove through self-supervised learning on natural language sentences. However, such embeddings can cover broader relationships than synonymous concepts, and may even include antonyms.

Our approach to building a semantic memory model embedding curates the knowledge graph relations to focus on semantic similarity. Specifically, focusing on the English language and their nouns, we assemble an initial list of semantically similar words by traversing the Wordnet thesaurus. We then curate the lists and uses this dataset to train a new embedding. We restricted to nouns both due to the use context of VLMs where we were applying this idea, and also due to the cost of curation by linguists. However, the process described below can be applied to other parts of speech.

**Development of a similarity list dataset**:

The initial similarity lists were obtained by directly traversing the Wordnet ontological tree gathering synonyms (called lemmas in WordNet) as well as hypernyms (generalizations) using the WU-Palmer similarity metric [24] which is given by:

$$sim(W_i, W_j) = 2 * \text{depth}[\text{lcs}(W_i, W_j)]/[\text{depth}(W_i) + \text{depth}(W_j)] \tag{1}$$

where where $\text{lcs}(W_i, W_j)$ is the least common ancestor of $W_i$ and $W_j$ and $\text{depth}(\cdot)$ stands for the depth of the concept in the ontology.

Without a constraint on the depth differential (2 in our case), and a reasonably high threshold, the WUP similarity score alone can reveal several false positives in association and lead to undesirable wider expansion of meanings, particularly for words closer to the root of the WordNet hierarchy. For example, with a 4 level depth differential for a word such as 'chair.n.05.chair' , the WUP similarity to the word 'device.n.01.device' is high (0.823) which is not synonymous. Further, the metric does not give a complete picture of semantic distance since domain-specific ontologies were constructed before their planned uses in textual embeddings. Also, due to the nature of the English language, the shortest-path distances between nodes or ontological depth differences do not have a uniform

implication of similarity across words. For example, synsets 'car.n.01' and 'van.n.01' are 16 apart in shortest path length, while 'car.n.01' and 'automobile.n.01' are only 1 apart. Conversely, terms that are not so close in meaning could also end up having a high score. For example, similarity metrics using depth differences can give similar scores for vastly different meaning words, e.g. (dog.n.01, giant panda.n.01) and (dog.n.01, hound.n.01) pairs have about the same WUP score of 0.86. .

Therefore, to normalize the notion of similarity, the initial lists produced by the automatic algorithm were curated by domain specialists. For WordNet, we used a team of 3 linguists from a nearby university to examine the similarity lists so that relationships other than similarity in meaning and sense were removed. Each linguists produced their own curated similarity lists. Triple consensus process was used to filter the lists so that those terms identified by all 3 linguists were retained in the final similarity list per anchor words. The original scores returned by the WUP metric were still retained for these pairs so that the linguists only filtered the irrelevant words from the lists but did not alter the WUP scores. For the WordNet ontology, we were able to address all valid nouns and their synonyms resulting in over 140,000 words. *Note that this vocabulary already exceeds the token vocabulary of most transformer models*. The whole curation process took over 1 year to complete.

**Developing the STE Embedding**:

We now develop a new textual embedding designed to capture the similarity relations reflected in these similarity lists such that only the word embeddings within a similarity list have high cosine similarity. Specifically, adopting the lemma notation of WordNet, we can characterize a word $W_i$ as:

$$W_i = <w_i, p_i, s_i, l_i> \qquad (2)$$

where $w_i$ is the multi-term word, $l_i \in \{\text{Synonym}(w_i)\}$ is a synonym, and $p_i \in \{n, a, v, r, s\}$ which stand for noun, adjective, verb, adverb, and adjective satellite respectively. Finally, $s_i$ stands for the sense of the word and is a number from $1$ to $n$.

Let $S_i$ be the set of semantically similar words for each anchor word $W_i$ as provided by the curated similarity lists. The semantic memory model uses these similarity lists to derive a neural encoding such that all words that mean the same or are semantically similar are pulled closer in the embedding. Thus pairs of anchor and target words from similarity lists are taken as positive examples, and all other pairings represent negative examples for the anchor class.

Specifically, given a fully-specified 4-tuple anchor word $W_i$, we encode it by a 1-hot encoding $O_i \in \{0, 1\}^{|V|}$, s.t. $\sum_{i=1}^{|V|} O_{ij} = 1$ as an input to the network where $V$ is the vocabulary. As a supervision label, we form a label vector in the real number space $Y_i = R^{|V|}$, s.t. $Y_{ij} = \text{sim}(W_i, W_j)$ iff $W_j \in S_i$ and 0 otherwise. Here $\text{sim}(W_i, W_j)$ is the similarity score returned from the similarity list generation. Thus each similarity list is characterized by a unique pattern label vector.

Architecture-wise, we design the semantic memory model as a unimodal, multi-label supervised contrastively-learned encoder. Specifically, the semantic memory model consists of an embedding layer to handle the large one hot vectors, a dense fully connected layer with ReLU activation for an encoder, and a decoder/projection network as another fully connected layer with ReLU activation, which is discarded after the learning, retaining only the encoder. The cosine similarity matrix between the encodings of the words is non-diagonal as shown in Figure 2c with the cells colored in green indicating the members of the similarity list as positive examples and the red colored cells representing the negative examples. In this case, for the candidate word "basket", the positive examples are "hamper, kreel, pannier" while "window, and table" are negative examples.

Specifically, the similarity between an anchor word $W_i$ at index $i$ in the vocabulary $\mathcal{V}$, and a candidate word $W_j \in S_i$ be captured by the contrastive loss per similarity list as:

$$\ell_{\text{contrast}}(S_i) = -\sum_{W_j \in S_i} \log \frac{\exp(z_i \cdot z_j / \tau)}{\sum_{a \in V} \exp(z_i \cdot z_a / \tau)} \qquad (3)$$

Here $z_i$ is the projected vector for word $W_i$ and $z_j$ is the projected vector similarly for $W_j \in S_i$. Finally, $z_a$ is the projected vector for any word $W_a$ either inside or outside the similarity list (i.e. ideally the entire vocabulary). In general, since the similarity lists are small in size, the number of negative samples to differentiate them need not take up the entire vocabulary $\mathcal{V}$, so smaller batch sizes could be used. $\tau$ is the temperature to weigh the contribution from similar vectors. Also, since

there are multiple such similarity lists, one for each vocabulary term, we can train them in sequential fashion through batching using a cumulative contrastive loss as $\mathcal{L}_{contrast} = \sum_{j}^{|\mathcal{V}|} \ell_{contrast}(S_j)$.

This type of non-diagonal similarity matrix formulation is unlike other self-supervised contrastively learned encoders such as CLIP[16]. Instead of a single positive image-text pair, we have several semantically similar words paired with an anchor word as positive examples. Further, we have additional supervision coming from the label given to a similarity list per word making it closer to supervised contrastive learning [10] but with multiple positive examples.

**Implementation Details:** Overall, the designed network architecture had the following parameters: input and output vector sizes= $142,989$, for various encoding size = $300, 1024, 2048, 4096$, and temperature= $0.05$ in the loss function. We used a batch size of $800$ and trained over a maximum of $10$ epochs or until the network error convergence was reached. We used the Adam optimizer for fast convergence with the learning rate as $0.001$. Two NVIDIA P100 GPUs with 16 GB were used for training and training took 5 hours. The network overall had 43,666,800 parameters (for encoding size of 300) and scaled accordingly for higher size encodings.

## 4   Developing the semantic alignment transform

The semantic alignment transform creates a linkage between the VLM and the semantic memory using the textual data that can be projected in both embedding spaces. Once the language concepts have been aligned, the image embeddings can utilize the transform of their nearest language concept to be also aligned with the semantic memory concepts, so that the textual phrases and image pairings of synonymous words are close to each other.

Modeling these desired transformations more formally, let $f_i(\cdot) : X_{\text{image}} \rightarrow R^{d_i}$ be the image encoder and $f_t(\cdot) : X_{\text{text}} \rightarrow R^{d_t}$ be the text encoder of a VLM model. Given a batch of $N_{\text{images}}$, $I_N = \{I_1, I_2, ..., I_N\}$ and $N_{\text{captions}}$, $T_N = \{T_1, T_2, ..., T_N\}$, we can project them into a common vector space $\mathcal{C} : R^d$, $C_t \in R^d$ of the VLM. In our notation, $C_i, C_t$ denote the vector representation in the VLM space for an image $I$, and a linguistic caption $T$, respectively. Consider two textual queries $q_1$ and $q_2$ which are synonymous of a query word $q$ (for e.g. "kreel", and "hamper" to "basket"). Let their projected vectors in the VLM space be $C_{q_1}, C_{q_2}, C_q$ and their nearest images be denoted by the vectors $C_{iq_1}$ and $C_{iq_2}$, respectively. Our goal is to design a semantic alignment transform $C'$ such that:

$$|C_j' - C_k'| < \delta, \text{where the indexes } j, k \in \{q, q_1, q_2, iq_1, iq_2\} \tag{4}$$

and $\delta$ is a small neighborhood so that both images corresponding to the vectors $C_{iq_1}$ and $C_{iq_2}$ are pulled up to either query $q_1$ or $q_2$.

Given a word $W_i$ projected in the original VLM space, its encoding in the semantic memory model can be denoted by $STE(W_i)$ such that

$$|STE(W_j) - STE(W_k)| < min(\gamma_{W_j}, \gamma_{W_k}), W_k \in \mathbf{S}_j \tag{5}$$

and $\delta$ where $W_j, W_k$ are words related by synonym relationship as defined in Wordnet, and $\mathbf{S}_j$ is the synonym list of word $W_j$ and $STE(W_j)$ is the semantic embedding of word $W_j$. $\gamma_{W_j}$ is the distance over which semantic similarity holds for $W_j$. Note that the distance $\gamma_{W_j}$ is a function of $W_j$, since some words have more synonyms than others.

The semantic alignment transform $(\Gamma_t(\cdot), \Gamma_i(\cdot))$ now projects the textual and image embeddings from VLM space such that

$$C_t' = \Gamma_t(C_t) \text{ and } C_i' = \Gamma_i(C_i) \tag{6}$$

where $\Gamma_i(\cdot) : R^{d_i} \rightarrow R^d$ and $\Gamma_t(\cdot) : R^{d_t} \rightarrow R^d$, where $R^d$ is the dimension of the STE space.

The alignment transform $\Gamma_t(\cdot)$ for text can be learned separately by mapping the embeddings of all words in a language from VLM space to the semantic memory space. However, we cannot train separately for $\Gamma_i(\cdot)$ because the STE embedding is only defined for text. To ensure that the image embeddings of synonymous words in VLM space are close to the synonymous words and their associated images in the STE space as well, we induce a transformation based on their nearest textual neighbors. Specifically, we can express $\Gamma_i$ as

$$\Gamma_i(C_i) = \Gamma_t(C_{t_i}) \text{   such that   } C_{t_i} = \arg\min_{t'} d(C_i, C_{t'}) \tag{7}$$

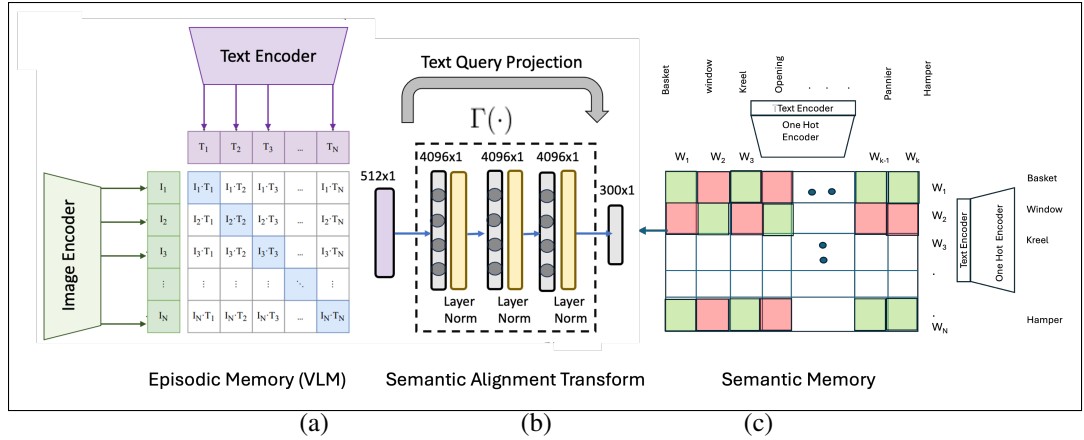

Figure 2: Architecture of SemCLIP demonstrating the various stages of creating the joint text-image embedding.

where $C_{t_i}$ is the nearest text to an image vector in the original VLM space in terms of distance $d(.)$: the cosine distance between the image and text vectors. This results in the image vectors aligning directly on top of the textual embedding vectors in the STE space.

**Learning the alignment transform**:

To train the alignment transform, we form a ground truth dataset of pairs of embeddings derived from a VLM model and the STE model for candidate words or phrases. While this method could be applied to any VLM model, in our work, we derived this mapping for the original CLIP model [16]. Unlike the STE embedding which was derived from WordNet, the alignment mapping used additionally, a much larger vocabulary of nearly 800,000 captions accumulated across datasets such as MS-COCO, Visual Genome and other collections.

For long captions, the correspondence was derived from the composed words in the caption and forming their average embeddings. For out-of-vocabulary words, we found the nearest match to their lexical variants in the vocabulary using an SBERT[17] encoding of the words/phrases. Since the nouns in the captions could be associated with multiple senses, an available word sense disambiguation (WSD) tool, ESC [1], was employed to resolve the sense of the constituent nouns before making the correspondence.

The alignment transform $\Gamma_t(\cdot)$ is a three layered Multi-layered Perceptron (MLP) with input size 512, output size 300 and intermediate layer width 4096 as shown in Figure 2. We use Layer Norm as the activation function. The network is trained using a Mean Squared Error (MSE) loss between the neural network outputs and the ground truth semantic embeddings. Equation 8 below captures the network details.

$$\underline{\text{Transform:}} \ \Gamma_t(\cdot) = \mathbf{FC}_3(\boldsymbol{\Phi}_{\text{relu}}(\mathbf{FC}_2(\boldsymbol{\Phi}_{\text{relu}}(\mathbf{FC}_1((\cdot)))))) \ \underline{\text{Loss:}} \ \mathcal{L} = ||\Gamma_t(C_t) - \mathbf{C}'_t||_2^2 \qquad (8)$$

To train the network, we use the ADAM optimizer with weight regularization (AdamW) and initial learning rate as 0.001. We train for a total of 200 epochs and use a batch size of 512. Along with the decrease in training loss, we calculate the retrieval errors (i.e. training fit using a nearest neighbors matching) after projection and observe less than 4 percent error in recovering the target semantic embeddings post-projection after training. Once $\Gamma_t(\cdot)$ is learned, the images were mapped using their nearest text embedding as described above.

By combining a VLM Model, the alignment transform, and the semantic memory model, the overall end-to-end architecture of SemCLIP is shown in Figure 2.

**Using SemCLIP for content stores**:

Using SemCLIP embedding, we can address the problem of semantic stability of retrieval in content stores as follows. All captions used to tag images along with the base vocabulary of the English language nouns are first projected into the SemCLIP space. During ingestion, the incoming images into the content store are projected into the new SemCLIP space using the transform of their nearest text embedding as given in Equation 7. Specifically, an incoming image file $I$ is mapped to a vector

Table 1: Illustration of synonym recognition across text embeddings.

| Embedding | # Queries | Synonyms in Top10 | %age synonyms covered |
|---|---|---|---|
| CLIP [15] | 71895 | 28070 | 49.27% |
| SBERT [17] | 71895 | 37888 | 52.7% |
| Ours | 71895 | 67309 | 87.7% |

248   $C_i' = \Gamma_t(C_{t_i})$ where $t_i = \arg\min_{t'} d(C_i, C_{t'})$ where $d$ is the cosine distance between the image and
249   text vectors in the original CLIP space $C$ as explained in Section 4. During retrieval, a new query $Q$
250   is projected into SemCLIP directly through the semantic text embedding of its composed entities as
251   $C_q'$. The nearest images to $Q$ are then retrieved within the neighborhood of $C_q'$ using cosine similarity
252   in the SemCLIP space.

## 5   Results

254   The SemCLIP model and its constituent embeddings were evaluated for semantic stability of image
255   retrieval on a variety of datasets as well as for many relevant downstream tasks such as image-to-text,
256   text-to-image retrieval, and text-to-text retrieval.

257   **Datasets**: We compare the performance of STE embedding on 13 benchmark datasets as listed in
258   Table 2. All datasets contain pairs of terms that are related in multiple ways ranging from synonyms
259   to antonyms, to part-of relations and have been used in previous evaluations. For the joint embedding,
260   we evaluated the performance of SemCLIP on 5 datasets, namely, Visual Genome [11], SUN [26],
261   CUB [21], AWA2 [25], MS-COCO, and Flicker30k. In each case, we retained all the labels and
262   the test image partition provided for these datasets. Each of the labels was processed using Spacy
263   to extract all noun entities. We then resolved their sense to give a 4-part notation for the nouns as
264   described earlier. The details of these datasets are described in Table 4 and Table 3.

265   **Comparison methods**: The semantic text embedding was compared to 4 popular word embedding
266   methods including, Word2Vec, Glove, BERT [3], and Path2Vec [12]. Since most image-text em-
267   beddings are variants of CLIP [15], our comparisons for SemCLIP included all popular variants,
268   namely, Open AI's original CLIP [15], OpenCLIP [15], NegCLIP [28], BLIP [13] and SigLIP[30].
269   In addition, we conducted ablation studies creating a variant of CLIP called PosCLIP by fine-tuning
270   CLIP directly with synonymous captions.

271   **Recognition of synonyms**: Since the STE model was trained with synonym similarity lists, we
272   expect a high overlap with synonyms in its topK retrieval in comparison to other textual and VLM
273   embeddings. To record this, we repeated the experiment described in Section 4 using SemCLIP
274   embedding and the result is shown as the last row in Table 1 indicating *nearly a doubling of*
275   *performance* over popular existing embeddings. Qualitatively, we found that due to the supervision
276   provided by the similarity lists, the neighborhoods in the STE embedding consist of only synonymous
277   or semantically related terms unlike other encodings like Word2Vec. For more quantitative results,
278   we evaluated the performance of STE embedding on 13 textual benchmarks shown in Table 2. The
279   resulting performance using the Spearman correlation coefficient to see the agreement of the similarity
280   ranked lists produced for each word in comparison to human ranked lists, is shown in that table. Our

Table 2: Illustration of comparative performance of semantic textual embeddings (STE) on benchmark datasets. The last column shows the STE result for the similar subet.

| Datasets | Original Word # | WordNet Filtered | Word2Vec | Glove | BERT | Path2Vec | STE | STE |
|---|---|---|---|---|---|---|---|---|
| EM_SIMLEX_SYNS | 297 | 297 | 0.285 | 0.240 | 0.145 | 0.301 | 0.265 | **0.570** |
| EN-MC-30 | 30 | 30 | **0.789** | 0.702 | 0.410 | **0.782** | 0.650 | 0.650 |
| EN-MEN-TR-3k | 3000 | 2657 | **0.776** | 0.743 | 0.310 | 0.366 | 0.257 | **0.780** |
| EN-MTurk-287 | 287 | 243 | 0.767 | 0.705 | 0.435 | 0.317 | 0.300 | **0.810** |
| EN-MTurk-771 | 771 | 771 | 0.671 | 0.649 | 0.335 | 0.404 | 0.466 | **0.760** |
| EN-RG-65 | 65 | 64 | 0.761 | 0.770 | 0.446 | 0.723 | 0.640 | **0.820** |
| EN-RW-STANFORD | 2034 | 910 | 0.492 | 0.341 | 0.226 | 0.194 | 0.217 | **0.590** |
| EN-SIMLEX | 666 | 666 | 0.452 | 0.397 | 0.233 | 0.505 | 0.398 | **0.670** |
| EN-WS-353-REL | 252 | 248 | **0.626** | 0.578 | 0.159 | 0.136 | 0 | 0 |
| EN-WS-353-SIM | 203 | 201 | 0.774 | 0.659 | 0.388 | 0.599 | **0.820** | **0.820** |
| EN-YP-130 | 130 | 43 | 0.542 | 0.545 | 0.326 | 0.029 | 0.426 | **0.660** |
| EW-WS-353-Syns | 99 | 98 | 0.507 | 0.507 | 0.366 | 0.616 | **0.655** | **0.655** |
| EN-WS-353-ALL | 352 | 348 | 0.694 | 0.607 | 0.256 | 0.406 | 0.303 | **0.720** |

Table 3: Results of average text-image retrieval overlap when querying using synonyms of nouns in the respective datasets. For each query, we use ten synonyms to estimate the image retrieval overlap

| Dataset | Images/Queries | Method | Overlap@1 | Overlap@5 | Overlap@10 | Overlap@50 |
|---|---|---|---|---|---|---|
| **Visual Genome** | 7794 / 14513 | **SemCLIP** | **0.551** | **0.532** | **0.517** | **0.523** |
| | | CLIP | 0.119 | 0.056 | 0.038 | 0.026 |
| | | OpenCLIP | 0.129 | 0.055 | 0.040 | 0.025 |
| | | BLIP | 0.119 | 0.056 | 0.037 | 0.024 |
| | | NegCLIP | 0.109 | 0.063 | 0.038 | 0.024 |
| **CUB** | 11788 / 200 | **SemCLIP** | **0.812** | **0.783** | **0.732** | **0.715** |
| | | CLIP | 0.118 | 0.072 | 0.062 | 0.053 |
| | | OpenCLIP | 0.149 | 0.079 | 0.062 | 0.053 |
| | | BLIP | 0.181 | 0.084 | 0.066 | 0.053 |
| | | NegCLIP | 0.119 | 0.078 | 0.066 | 0.055 |
| **SUN** | 16657 / 567 | **SemCLIP** | **0.554** | **0.531** | **0.523** | **0.511** |
| | | CLIP | 0.092 | 0.048 | 0.034 | 0.025 |
| | | OpenCLIP | 0.070 | 0.039 | 0.028 | 0.021 |
| | | BLIP | 0.091 | 0.05 | 0.035 | 0.025 |
| | | NegCLIP | 0.095 | 0.045 | 0.032 | 0.023 |
| **AWA2** | 6985 / 10 | **SemCLIP** | **0.751** | **0.723** | **0.702** | **0.715** |
| | | CLIP | 0.086 | 0.051 | 0.038 | 0.028 |
| | | OpenCLIP | 0.139 | 0.067 | 0.046 | 0.032 |
| | | BLIP | 0.139 | 0.057 | 0.039 | 0.028 |
| | | NegCLIP | 0.101 | 0.046 | 0.034 | 0.025 |

method was expected to perform worse on the datasets where the relations are antonyms or other forms of relations besides meaning similarity, but should perform better when limited to the meaning-wise similar pairs in these benchmark datasets. As seen in Table 2, it significantly outperforms other embeddings in the case of the EN-WS-353-SIM dataset which focuses on similarity relations. If we restrict the analysis to only the similar words in all datasets, our method outperforms all other methods as shown in the last column. Finally, for datasets such as EN-WS-353-REL which capture antonyms and other relationships besides synonyms, our performance is the least, which is also a good result indicating it is able to focus on similarity relations only. Note that the values in Table 2 are Spearman correlation coefficient where the values above 0.7 indicate strong correlation which our method achieves for most datasets.

**Evaluating stability in retrieval**: We evaluated the stability of retrieval by measuring the overlap in the image lists returned in response to queries and their synonym variants. Specifically, we extracted nouns from each of the captions covered by the test partitions of the respective datasets. All text to image retrieval used a common prompt of "A photo of " before each noun flagged in a caption. We then recorded the pairwise overlap of the top K lists returned for a caption with the top K lists of images returned from their synonym replacements. The overlap was averaged across the synonym replacements to serve as a measure of the stability of retrieval. The experiments were performed for all CLIP variants. The result is shown in Table 3. As can be seen, by projecting the synonymous phrases to the SemCLIP embedding, the list of images returned show far higher overlap in SemCLIP in comparison to other CLIP variants.

Due to the projection of synonymous phrases and their associated embedding close together, we expect an increase in the precision and recall for general text-to-image retrieval as well. We evaluated this using the popular measures of NDCG and mean average precision (MAP). To keep the comparison fair, all ground truth labels of images were augmented with their synonym equivalents. For example, images labeled with 'clock frame' were also augmented with the label 'frame/clock' from the same caption set as both these labels share the same entities and would be represented by the same average vector in SemCLIP space. For each dataset, our method achieves the highest NDCG@K as well as MAP across various datasets as shown in Table 4 (under the columns "t2i") except for AWA2 which had the fewest labels.

**Evaluating image-to-text retrieval:** The image-to-text retrieval experiments results also showed similar performance as shown in Table 4 under the columns "i2t". Note that when there are large number of captions (visual genome, Flickr30k), our method's performance is best seen due to the capturing of semantics of multiple noun phrases in the average vector embeddings used in the transformation. The COCO and Flickr30K labels were not used for training the alignment mapping of SemCLIP.

Table 4: Comparisons of text-to-image (t2i) and image-to-text (i2t) retrieval performance with different models.

| Dataset | Images / Labels | Model | t2i: NDCG / mAP / Recall (@10) | i2t: NDCG / mAP / Recall (@10) |
|---|---|---|---|---|
| **Visual Genome** | 7794 / 14513 | **SemCLIP** | **0.192 / 0.172** | **0.254 / 0.185** |
| | | CLIP | 0.053 / 0.060 / 0.065 | 0.050 / 0.129 / 0.028 |
| | | OpenCLIP | 0.066 / 0.075 / 0.081 | 0.061 / 0.159 / 0.035 |
| | | BLIP | 0.072 / 0.081 / 0.088 | 0.069 / 0.167 / 0.039 |
| | | NegCLIP | 0.063 / 0.072 / 0.077 | 0.060 / 0.150 / 0.035 |
| | | PosCLIP | 0.074 / 0.084 / 0.091 | 0.060 / 0.146 / 0.034 |
| **CUB** | 11788 / 200 | **SemCLIP** | **0.721 / 0.845** | **0.891 / 0.812** |
| | | CLIP | 0.513 / 0.621 / 0.084 | 0.619 / 0.554 / 0.826 |
| | | OpenCLIP | 0.669 / 0.744 / 0.111 | 0.777 / 0.726 / 0.935 |
| | | BLIP | 0.204 / 0.341 / 0.033 | 0.326 / 0.260 / 0.543 |
| | | NegCLIP | 0.406 / 0.535 / 0.065 | 0.472 / 0.403 / 0.694 |
| | | PosCLIP | 0.488 / 0.580 / 0.080 | 0.553 / 0.479 / 0.788 |
| **SUN** | 16657 / 567 | **SemCLIP** | **0.686 / 0.712** | **0.810 / 0.671** |
| | | CLIP | 0.414 / 0.562 / 0.191 | 0.458 / 0.415 / 0.595 |
| | | OpenCLIP | 0.549 / 0.664 / 0.260 | 0.514 / 0.476 / 0.634 |
| | | BLIP | 0.413 / 0.535 / 0.194 | 0.426 / 0.384 / 0.557 |
| | | NegCLIP | 0.429 / 0.553 / 0.201 | 0.425 / 0.380 / 0.567 |
| | | PosCLIP | 0.463 / 0.602 / 0.214 | 0.445 / 0.399 / 0.589 |
| **AWA2** | 6985 / 10 | **SemCLIP** | 0.967 / 0.987 | 0.995 / 0.989 |
| | | CLIP | 0.993 / 0.999 / 0.016 | 0.992 / 0.989 / 1.000 |
| | | OpenCLIP | **1.000 / 1.000** / 0.016 | **0.994 / 0.991** / 1.000 |
| | | BLIP | **1.000 / 1.000** / 0.016 | 0.991 / 0.988 / 1.000 |
| | | NegCLIP | **1.000 / 1.000** / 0.016 | 0.987 / 0.982 / 1.000 |
| | | PosCLIP | **1.000 / 1.000** / 0.016 | 0.983 / 0.977 / 1.000 |
| **COCO** | 5000 / 80 | **SemCLIP** | **0.940** / 0.91 | **0.895 / 0.923** |
| | | CLIP | 0.810 / 0.869 / 0.081 | 0.706 / 0.771 / 0.745 |
| | | OpenCLIP | 0.866 / 0.926 / 0.087 | 0.756 / 0.788 / 0.799 |
| | | BLIP | 0.897 / 0.942 / 0.089 | 0.774 / 0.852 / 0.781 |
| | | NegCLIP | 0.834 / 0.892 / 0.083 | 0.766 / 0.797 / 0.808 |
| | | PosCLIP | 0.919 / 0.946 / 0.088 | 0.807 / 0.834 / 0.843 |
| **Flickr30k** | 31014 / 158391 | **SemCLIP** | **0.571 / 0.523** | **0.580 / 0.572** |
| | | CLIP | 0.354 / 0.306 / 0.510 | 0.314 / 0.430 / 0.320 |
| | | OpenCLIP | 0.427 / 0.377 / 0.588 | 0.376 / 0.489 / 0.382 |
| | | BLIP | 0.562 / 0.510 / 0.725 | 0.475 / 0.587 / 0.480 |
| | | NegCLIP | 0.425 / 0.373 / 0.591 | 0.354 / 0.465 / 0.360 |
| | | PosCLIP | 0.323 / 0.279 / 0.468 | 0.273 / 0.367 / 0.283 |

**Performance on classification:** We evaluated SemCLIP also for the standard task of classification using the predicted labels. On the ImageNet classification, our zero shot classification accuracy for SemCLIP was at 88.3% in comparison to CLIP at 84.2%.

**Ablation studies**: To explore the value of using the semantic alignment to a semantic memory model, we conducted an ablation study in which we fine-tuned CLIP on the visual genome dataset by directly providing the synonymous captions as positive examples and using binary cross-entropy loss to cover the multiple positive examples. The performance of the resulting model called POSCLIP can be seen in Table 4 which are not as impressive as doing an explicit transformation of the terms into the SemCLIP space, indicating the lack of similarity in the original textual embeddings of synonymous terms.

# 6 Conclusions

In this paper, we offered new approach to semantically enriching VLM models by drawing inspiration from the linkages of episodic and semantic memory in the brain. A simple model of semantic memory was developed covering all nouns in the English language. A linkage transform was developed to map from VLM space to semantic memory space. The resulting VLM model was shown to outperform on multiple tasks against multiple datasets. Performance variation was still seen across datasets due to their incomplete ground truth labeling, but SemCLIP was found to be better for larger vocabularies due to the better handling of synonymous terms. Future work will extend this paradigm to cover other parts of speech, and better address out-of-vocabulary words and word-sense disambiguation issues.

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
