# A Appendix

In this appendix, we present additional details to support the conclusions of our paper. The paper itself is self-contained and covered many aspects of our SemCLIP model generation. However, there were several additional experiments done that can throw some more light into the work done for the paper. These are captured in the sections below.

## A.1 Handling arbitrary text captions through average vectors

In this section, we present the rationale for representing arbitrary captions through the average embedding formed from their composed words. The composed words are derived from the Wordnet thesaurus. The following discussion is applicable other textual embeddings besides our STE model. The notations used here are the same as in Section 4.

The rationale for the average vector approach comes from two sources. First, the VLMs are able to retrieve relevant images to textual queries even when they are expressed simply as a collection of grammatical entities. Consider a full caption: "There is a table in the middle of the room". The composed non-stop and useful words in this sentence can be easily extracted through standard NLP methods as "table", "middle", "room". If we ignore the preposition, and focus only on nouns, then the composed nouns are "table" and "room". By using the average CLIP text vectors of these nouns, the images retrieved are roughly similar to those retrieved by the use of the full caption as seen from Figure 1(e)-(g) where relevant matches are obtained from both a full fledged phrase shown (Figure 1(e)) as well as when broken down into a set of nouns only ("table", "room") (Figure 1(f)) or in any order ("room", "table") in Figure 1(g). Thus it seems plausible to represent an arbitrary caption in terms of its essential composed words and in particular, the constituent nouns depicting objects.

Our experimental validation also showed that replacing queries by the average vectors of embeddings from their composed nouns gives similar retrieval performance. Figure 3(a) and (b) shows the results of finding similar captions in the CLIP embedding space for the entire set of 83404 captions in the Visual Genome dataset [12] based on their average vectors. As can be seen from Figure 3(a), the original caption was the nearest vector for 90% of the average vectors with an average cosine similarity of 0.958. We also repeated this in an image to text similarity experiment using the original captions vectors and their average versions on all the 7554 images of the test partition of the Visual Genome dataset [12]. As can be seen from the results in Table in Figure 3(b), the performance using average vectors is comparable to the performance with the original captions.

In fact, we can make the following proposition.

**Proposition-1:** The vector representation $C_q$ of a query $Q$ in a VLM space $C$ can be approximated by the average vector $C_{\text{avg}} = \frac{\sum_j C_{ej}}{N_q}$ where $C_{ej}$ is the vector representation of the entity $e_j$ in the VLM space $C$ and $N_q$ are the number of entities composing the query $Q$.

A second rationale comes from the fact that if we develop a semantic text embedding for words that preserve the synonymous relationship of individual grammatical elements, e.g. nouns, we can expect their enclosing synonymous queries to preserve their relationships in the projected space as well.

**Corollary-1:** Given pairs of synonymous queries $Q_1, Q_2$ represented by their average vectors $C'_{\text{avg1}}, C'_{\text{avg2}}$ formed from $C'_{eq11}, ..C'_{eq1k}$ and $C'_{eq21}, ..C'_{eq2k}$, if $|C'_{eq1l} - C'_{eq2l}| < \delta$ for all $C'_{eql}$ then $|C'_{\text{avg1}} - C'_{\text{avg2}}| < \delta$. This follows directly from vector averaging rules.

## A.2 More details on semantic text embedding learning

In this section, we provide further details on our semantic text embedding.

### A.2.1 Representing word senses in STE embedding

We note that unlike other word embeddings which either have a unique embedding (e.g. Word2Vec) or variable embeddings based on use context (e.g. BERT), our representations of a word in STE embedding are only as many as the senses in which the word occurs in the language. Consider an example word 'lemon' which has 5 senses, even though not all 5 of them begin with the word lemon in the synset definitions of Wordnet. Lemon is a lemma (in Wordnet, the synonyms are captured

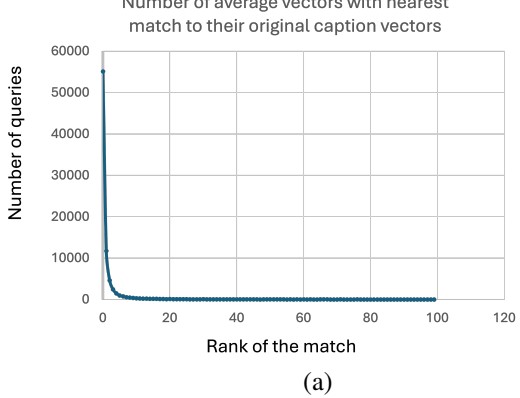

| Top K | %match using full caption | % match using average vector |
|---|---|---|
| 1 | 4.38% | 3.34% |
| 5 | 11.70% | 10.26% |
| 10 | 16.35% | 14.90% |
| 50 | 32.04% | 30.52% |

| (a) | (b) |
|---|---|

Figure 3: (a) Illustration of text-to-text retrieval using captions approximated by average vectors. (b) Illustration of image-to-text retrieval using full caption vectors and approximation by average vectors.

as lemmas) in 'lemon.n.01.lemon' using the synset 'lemon.n.01' which has the meaning "'yellow oval fruit with juicy acidic flesh' . Lemon is also a lemma or synonym of the word 'gamboge' in the form 'gamboge.n.01.lemon' whose synset 'gamboge.n.01' has the meaning 'a gum resin used as a yellow pigment and a purgative' so that the reference to lemon here is for its color. In our STE embedding, 'lemon.n.01.lemon' and 'gamboge.n.01.lemon' are two different embeddings and will have two different similarity sets, the former grouping lemon variants of the fruit, while the latter referring to resins and gums.

### A.2.2 Curation of similarity lists by linguists

The curation process used for cleaning the similarity lists automatically traversed in Wordnet removed many of the spurious similarities. Table 5 shows an example for one such similarity list for the word mutual_fund.

Table 5: Illustration of the similarity curation process with linguists. The initial similarity list for the concept Fund.n.01.mutual_fund had over 20 nouns which were filtered to 10 by the linguists as shown in the third column.

| Initial Similarity List I | Initial similarity list II (Contd.) | Retained after curation (11) |
|---|---|---|
| budget.n.01.budget 0.94
civil_list.n.01.civil_list 0.89
demand_deposit.n.01.demand_deposit 0.89
deposit.n.04.deposit 0.94
exchange_traded_fund.n.01.exchange_traded_fund 0.89
fund.n.01.fund 1.00
fund.n.01.monetary_fund 1.00
index_fund.n.01.index_fund 0.89
medium_of_exchange.n.01.medium_of_exchange 0.86
money.n.01.money 0.93 | mutual_fund.n.01.mutual_fund 0.94
operating_budget.n.01.operating_budget 0.89
pension_fund.n.01.superannuation_fund 0.94
petty_cash.n.01.petty_cash 0.94
revolving_fund.n.01.revolving_fund 0.94
savings.n.01.nest_egg 0.94
sinking_fund.n.01.sinking_fund 0.94
slush_fund.n.01.slush_fund 0.94
trust_fund.n.01.trust_fund 0.94
war_chest.n.01.war_chest 0.94 | exchange_traded_fund.n.01.exchange_traded_fund 0.89
fund.n.01.fund 1.00
fund.n.01.monetary_fund 1.00
index_fund.n.01.index_fund 0.89
money.n.01.money 0.93
mutual_fund.n.01.mutual_fund 0.94
pension_fund.n.01.superannuation_fund 0.94
revolving_fund.n.01.revolving_fund 0.94
sinking_fund.n.01.sinking_fund 0.94
slush_fund.n.01.slush_fund 0.94
trust_fund.n.01.trust_fund 0.94 |

### A.2.3 STE embedding is more than looking up Wordnet for synonyms

The semantic text embedding developed covers many more synonymous relationships between nouns than explicitly captured through direct synonyms, hypernyms and hyponyms of Wordnet although they are recoverable from Wordnet through careful navigation. Table 6 lists a few examples indicating the expanded list produced by searching in our semantic text embedding.

### A.2.4 Qualitative comparison of STE with Word2Vec

The STE embedding by way of training with similarity lists ensures that the top matches all captures the synonymous relations in comparison to other textual encodings such as Word2Vec. This is illustrated in Table 7.

Table 6: Illustration of synonym expansion through SemCLIP text embedding to show that the retrieved synonyms are more than what can be obtained by Wordnet alone indicating the additional value of SemCLIP for other downstream use cases requiring semantic text analysis.

| Query | Search in Wordnet | Search in Semantic Embedding |
|---|---|---|
| 'hood | {'hood', vicinity} | {'hood', 'proximity', 'gold_coast', 'locality', 'neighbourhood', 'neck_of_the_woods', 'neighborhood', 'place', 'section', 'vicinity'} |
| abdominal cavity | {'abdominal_cavity', 'cavity', 'abdomen'} | {'pit_of_the_stomach', 'orbital_cavity', 'glenoid_cavity', 'cavity', 'axillary_fossa', 'abdomen', 'orbit', 'cavum', 'abdominal_cavity', 'bodily_cavity'} |
| erosion | {'erosion', 'ablation'} | {'erosion', 'deflation', 'wearing_away', 'ablation', 'detrition', 'eroding', 'abrasion', 'attrition', 'eating_away', 'wearing'} |
| dorm room | {'dorm_room', 'dormitory_room', 'dormitory', 'bedroom'} | {'dormitory_room', 'chamber', 'sleeping_accommodation', 'master_bedroom', 'dormitory', 'dorm_room', 'sleeping_room', 'bedroom', 'bedchamber', 'guestroom'} |

Table 7: Sample top 10 results below show the quality of matches from Word2Vec versus our embedding where non-synonyms can be seen in the Word2Vec list.

| Query | Top 10 Results |
|---|---|
| van (Word2Vec) | 'car', 'parking lot', 'parking meter', 'friend', 'back', 'suv', 'two', 'vehicle', 'street sign', 'front' |
| tree trunk (Word2Vec) | 'tree trunk', 'trunk', 'tree', 'tree branch', 'pole', 'pine tree', 'ski pole', 'christmas tree', 'telephone pole', 'line' |
| van (STE) | 'van.n.01', 'car.n.01', 'sport utility.n.01', 'jeep.n.01', 'cab.n.01', 'minivan.n.01', 'sedan.n.01', 'automotive vehicle.n.01', 'motor vehicle.n.01', 'delivery van.n.01' |
| tree trunk (STE) | 'tree trunk.n.01', 'plant organ.n.01', 'trunk.n.01', 'stalk.n.02', 'bole.n.01', 'stem.n.02', 'wood.n.01', 'pole.n.01', 'structure.n.01' |

## A.3 More details on alignment transform learning

### A.3.1 Captions used for training the alignment transform

Table 8 records the details of the entity breakdown process used for analyzing the captions in the various datasets used for training our transformation mapping. A total of 799702 captions were used that included the WordNet nouns as well.

Table 8: Details of captions from various datasets used for training the transformation mapping.

| Dataset | Captions | Captions with Noun | Captions with Entity | Caption with Noun Phrase | Caption with Tokens |
|---|---|---|---|---|---|
| MS-COCO | 568456 | 568372 | 96956 | 568414 | 568456 |
| Visual Genome | 83404 | 69410 | 14157 | 76121 | 83404 |
| CUB | 200 | 58 | 46 | 109 | 200 |
| SUN | 567 | 239 | 105 | 357 | 567 |
| AWA | 50 | 22 | 9 | 39 | 50 |
| Wordnet | | 147025 | | | |
| **Total** | | | | | 799702 |

### A.3.2 Word sense disambiguation used

We used the word-sense disambiguation tool ESC[2] for parsing the captions to find the correct sense of the constituent nouns in the caption. Specifically, for each noun in a caption, we used the noun as the target word and the caption as the context, and the output was the sense of the noun in the

Table 9: Illustration of positive examples generation for training the PosCLIP model. Less sensible captions generated by synonym substitutions are filtered by SBERT.

| Original phrase | pony toy |
|---|---|
| Synonyms of each word | toy → plaything, water pistol, hobby, rocking horse, slingshot, catapult |
| | pony → cayuse, Indian pony, horse, Equus caballus |
| All possible combinations | cayuse plaything, Indian pony plaything, horse plaything, Equus caballus plaything, pony plaything, cayuse water pistol, Indian pony water pistol, horse water pistol, Equus caballus water pistol, pony water pistol, cayuse hobby, Indian pony hobby, horse hobby, Equus caballus hobby, pony hobby, cayuse rocking horse, Indian pony rocking horse, horse rocking horse, Equus caballus rocking horse, pony rocking horse, cayuse slingshot, Indian pony slingshot, horse slingshot, Equus caballus slingshot, pony slingshot, cayuse catapult, Indian pony catapult, horse catapult, Equus caballus catapult, pony catapult, cayuse toy, Indian pony toy, horse toy, Equus caballus toy, pony toy |
| With SBERT cos sim > 0.8 | pony toy, pony plaything, Indian pony toy, horse toy |

4-part notation mentioned given in Eqn (2). We used the model checkpoint [1] provided by the authors of ESC to do the sense disambiguation. This model, given a target noun, and a sentence, picks the best sense of the noun in terms of Wordnet synsets. For example, in a phrase such as "Lion and giraffe in separated enclosures at the zoo", and the target noun "lion" it disambiguates among the three senses of nouns in Wordnet and correctly picks the sense 'lion.n.01' (lion as an animal) against 'lion.n.02' (celebrity) or 'lion.n.03' (leo sign of the zodiac). Among the WSD tools available in literature, this was the best performing with an accuracy of 80% indicating this is still a challenging research problem. For example, in the sentence "an old fashioned colonial dining room hutch and an anniversary clock on a shelf on the wall" and using the target noun as 'hutch', it maps to the synset 'hovel.n.01' which means a crude shelter and not the furniture as intended here.

## A.4 Generating fine-tuned CLIP model using synonyms

The ablation study showed that fine-tuning CLIP using synonyms of words or creating synonymous variants of captions as positive examples does not offer the same advantages as projecting CLIP embeddings to STE embeddings. While synonyms of single word captions can be directly looked up in Wordnet, generating synonymous phrases for arbitrary captions posed challenges since not every substitution resulted in a meaningful caption. Table 9 shows this through the manipulation of a single caption named "pony toy". Not all combinations generated by substituting each noun in the phrase by its synonym is a valid combination or even meaningful phrase in the English language.

## A.5 Using SemCLIP embedding for deploying in cloud vector stores

We can use SemCLIP image-text embedding for deployment in any vector store as follows. We initialize the textual embeddings of the vector store with all nouns and text captions used during training to serve as initial vocabulary. Any new text caption acquired during subsequent deployment can be added as an average vector formed from its constituent nouns. An incoming image file $I$ is mapped to a vector $C_i^{'} = \Gamma_t(C_{t_i})$ where $t_i = \arg\min_{t'} d(C_i, C_{t'})$ where $d$ is the cosine distance between the image and text vectors in the original CLIP space $C$ as explained in Section 4. A new query $Q$ is projected into SemCLIP directly through the semantic text embedding of its composed entities as $C_q^{'}$. The nearest images to $Q$ are retrieved within the neighborhood of $C_q^{'}$ using cosine similarity in the SemCLIP space.

---

[1] https://github.com/SapienzaNLP/esc