# OpenReview forum: "SemCLIP: A Semantic Memory-Aligned Vision Language Model"
_NeurIPS.cc/2025/Workshop/UniReps — UniReps2025 oral_

### Official Review · Reviewer_6BM3 · 2025-09-15
**Semantic memory-aligned VLM that boosts synonym stability but with limited scope**

**Confidence:** 5

**Review:**

Summary
This paper introduces SemCLIP, a vision-language model that integrates a semantic memory inspired by human cognition. The authors construct a Semantic Text Embedding (STE) using curated WordNet similarity lists and supervised contrastive learning, then design a semantic alignment transform to map VLM embeddings into this semantic memory space. The approach stabilizes retrieval across synonymous queries and enhances semantic consistency without retraining large VLMs.

Strengths
1. Drawing on episodic-semantic memory interaction provides a fresh perspective for addressing synonym instability in VLMs.

2. The curated WordNet-based synonym lists (over 140K nouns, 600K+ pairs) are a valuable open-source contribution for NLP and VLM tasks. Also, the paper reports strong improvements in retrieval stability and synonym recognition (e.g., 87.7% coverage vs ~50% for CLIP/SBERT), with consistent boosts on standard datasets.

3. The semantic alignment avoids large-scale retraining of VLMs, making the approach computationally efficient and practical. Also, the authors have done a comprehensive evaluation. Results span text-to-image, image-to-text, and zero-shot classification, with both ablation studies and multiple baselines.

4. The paper demonstrates improvements across a wide range of benchmarks (Visual Genome, SUN, CUB, COCO, Flickr30K, etc.), showing that the semantic alignment strategy generalizes well across both fine-grained and large-scale datasets .

5. The ablation study (PosCLIP vs. SemCLIP) clearly shows that simply fine-tuning CLIP on synonym pairs is insufficient, and that explicit semantic alignment yields significantly better retrieval stability .

Areas for improvement
1. Restricting semantic memory to nouns narrows applicability; verbs/adjectives are equally important for richer queries.

2. Reliance on linguist-curated similarity lists (over one year of work) raises scalability concerns for other languages or domains.

3. Word-sense disambiguation remains fragile; handling polysemy in real-world captions may reduce robustness. Also while retrieval stability is well-studied, downstream generative tasks (e.g., captioning, reasoning) are not evaluated.


Suggestions to authors
1. Extend the experiment beyond nouns. Explore verbs, adjectives, and multi-word expressions to capture richer semantics and improve coverage in natural queries.
2. Reduce reliance on manual curation by leveraging large language models, crowdsourcing, or semi-supervised expansion of synonym lists.
3. Incorporate stronger word sense disambiguation mechanisms (beyond ESC) to better manage context-dependent meanings.

**Score:**

4

**Topic Fit:**

3

---

### Official Review · Reviewer_s38m · 2025-09-15
**SemCLIP: Aligning VLMs to a Curated “Semantic Memory” for Synonym-Stable Retrieval**

**Confidence:** 3

**Review:**

**Summary**
The paper proposes SemCLIP, which learns (i) a supervised textual embedding (“semantic memory”) from linguist-curated WordNet similarity lists and (ii) a learned alignment transform that maps CLIP text embeddings into this semantic space. Image embeddings are then aligned indirectly via their nearest text in CLIP space. The result aims to make retrieval robust to synonym swaps and to improve text–image and image–text retrieval metrics across several datasets. The manuscript is well-motivated, reports strong gains on synonym recognition and retrieval stability, and releases a sizable curated resource; however, there are concerns about novelty relative to knowledge-graph-augmented VLMs, possible evaluation circularity, reliance on nearest-text proxies for images, and incomplete ablations/reporting.

**Pros**

- Novel and Well-Motivated Approach: The analogy to episodic and semantic memory provides a strong conceptual foundation. Moving beyond simple fine-tuning and instead learning an explicit mapping ($\Gamma(\cdot)$) to a structured semantic space is a novel and powerful idea for injecting external knowledge into pre-trained models.

- Strong and Comprehensive Evaluation: The authors evaluate their model on a wide range of tasks, including synonym recognition, semantic similarity benchmarks (Table 2), retrieval stability (Table 3), and standard image/text retrieval (Table 4). The inclusion of an ablation study (PosCLIP) effectively demonstrates that their alignment approach is superior to simply fine-tuning the VLM with synonymous captions.

- Careful scope statement: authors acknowledge that focusing on similarity de-emphasizes other relations (antonymy, meronymy), and the numbers reflect that.

**Cons**

- Metric definitions. “Overlap@K” is not formally defined. Please define.
- The claimed zero-shot ImageNet jump to 88.3% (vs CLIP 84.2%) is large; please specify the exact CLIP backbone
- Define all datasets, splits, and caption processing exactly

**Originality**
Moderate. The combination of a curated, supervised “semantic memory” with a learned text-space alignment into which images are snapped via nearest text is a neat operationalization; however, aligning VLMs to external semantic structures (WordNet/ConceptNet/KGs) is a well-traveled path. The novelty claim would be stronger with head-to-head comparisons to KG-augmented CLIP and with principled mapping baselines (orthogonal Procrustes/CCA).

**Significance**
Practically meaningful for enterprise retrieval and tagging: synonym stability matters, and the proposed approach is easy to bolt on. Research-wise, significance depends on whether improvements persist under hard, non-synonym tests (compositionality, attributes, relations), how robust the method is to WSD/OOV, and whether a learned image mapping closes the remaining gap.

**Clarity**
Generally readable with helpful math. Needs tighter definitions, explicit data splits, and stronger presentation of failure modes and error bars. Release of the curated lists is excellent; please also release code and the trained alignment model.

**Score:**

4

**Topic Fit:**

3

---

### Official Review · Reviewer_gQT8 · 2025-09-15
**SemCLIP: A Semantic Memory-Aligned Vision Language Model**

**Confidence:** 3

**Review:**

This paper introduces SemCLIP, a semantic vision-language model that links a vision-language model to a semantic memory system via an alignment transform, inspired by how the brain connects episodic and semantic memory. By incorporating semantic similarity between object-naming nouns, SemCLIP achieves more stable concept representations and outperforms existing models on semantic similarity and retrieval tasks without large-scale retraining.

Strengths:

1.Built a new text embedding (STE) that links nouns by their semantic similarity.

2.Added a semantic alignment transform to connect VLM embeddings with this semantic memory.

3.Released a huge dataset of word similarity lists and synonym pairs for the community.

Weakness：

1.How was the similarity list dataset constructed? Some symbols in the paper are not standardized, making it difficult to understand.

2.The results for SemCLIP in Table 4 are ambiguous and cannot be directly interpreted from the table. Which specific one should be labeled?

3.How does SemCLIP's comparison method differ from some open-source approaches in terms of training data? The experiments include some that appear to be conducted on a large scale. How can SemCLIP's advantages be demonstrated from the results?

**Score:**

3

**Topic Fit:**

2